# Passive Smoking Exposure in Living Environments Reduces Cognitive Function: A Prospective Cohort Study in Older Adults

**DOI:** 10.3390/ijerph17041402

**Published:** 2020-02-21

**Authors:** Fan He, Tian Li, Junfen Lin, Fudong Li, Yujia Zhai, Tao Zhang, Xue Gu, Genming Zhao

**Affiliations:** 1Department of Epidemiology, School of Public Health, Fudan University, Shanghai 200032, China; fhe@cdc.zj.cn; 2Zhejiang Provincial Center for Disease Control and Prevention, Hangzhou 310051, China; zjlinjunfen@163.com (J.L.); fdli@cdc.zj.cn (F.L.); yjzhai@cdc.zj.cn (Y.Z.); tzhang@cdc.zj.cn (T.Z.); xgu@cdc.zj.cn (X.G.); 3Mailman School of Public Health, Columbia University Medical Center, New York, NY 10032, USA; tl2882@cumc.columbia.edu

**Keywords:** passive smoking, cognitive impairment, longitudinal study, hazard ratio, dementia, older adults, aging

## Abstract

There is currently no consensus regarding the effects of passive smoking exposure on cognitive function in older adults. We evaluated 7000 permanent residents from six regions within Zhejiang Province, China, aged ≥60 years, without cognitive impairment at baseline and during follow-up examinations for two years. The Chinese version of the Mini-Mental State Examination was used to assess the participants’ cognitive function. Multivariate regression analyses were carried out to calculate the adjusted relative risks (RRs) as measures of the association between passive smoking exposure and cognitive impairment after adjusting for potential confounders. The results showed an association between passive smoking exposure in the living environment and increased risk of cognitive impairment (RR: 1.16; 95% confidence interval (CI): 1.01–1.35). No dose–response relationship between the cumulative dose of passive smoking exposure (days) and cognitive impairment was observed. The results of stratified analyses suggested a harmful effect of passive smoking exposure on cognitive function in non-smokers (RR: 1.24; 95% CI: 1.06–1.46), but not in smokers (RR: 1.11; 95% CI: 0.71–1.92). Therefore, passive smoking exposure increased the risk of cognitive impairment in older adults, especially non-smokers. More effective measures to restrict smoking in the living environment should be developed and implemented.

## 1. Introduction

With increasing population aging worldwide, cognitive impairment is becoming an increasingly common and important health-care challenge for older people, with negative effects on the activities of daily living of older people and burdens on both their families and society. Therefore, identification of the factors associated with cognitive impairment is imperative to reduce risks in older people.

Passive smoking, also called “secondhand smoke,” refers to the mixed smoke released from the tobacco products of other smokers. Secondhand smoke contains more than 7000 harmful chemicals and dozens of carcinogens [1,2]. Exposure to passive smoking can increase the risk of developing or dying from diseases such as cancer (particularly lung and breast cancer) and cardiovascular diseases (e.g., stroke, angina, and hypertension), as well as cognitive impairment [3,4,5]. Positive correlations between tobacco exposure and cognitive impairment, dementia, and other neurodegenerative diseases have also been reported [6,7]. However, some studies have suggested a protective role of nicotine against cognitive impairment [8,9]. Therefore, in this prospective cohort study, we recruited nearly 10,000 older adults from six counties in Zhejiang Province of China, in 2014, to further explore the relationship between passive smoking exposure and cognitive impairment in older adults.

## 2. Materials and Methods

### 2.1. Participants

The participants were enrolled in the Zhejiang Major Public Health Surveillance Program (ZJMPHS), a prospective study of health issues among older adults that started in 2014, with follow-up investigations conducted in 2015 and 2016. Detailed information on the program has been provided previously [10,11]. At baseline (2014), the ZJMPHS survey included 9353 permanent residents aged ≥60 years, from six counties in Zhejiang Province. A total of 7947 participants with no cognitive impairment at baseline were followed up for two years. However, 447 (5.6%) participants died, and 500 (6.3%) were lost to follow-up. The remaining 7000 participants who completed the baseline and follow-up investigations were included in the analysis. Written informed consent was obtained from each participant. The program was approved by the Ethics Committee of the Zhejiang Provincial Center for Disease Control and Prevention (2018-50).

### 2.2. Assessment of Passive Smoking Exposure in Living Environments

Passive smoking exposure was assessed by asking the participants, “Has anyone living with you smoked in recent years?” Participants who answered “yes” were further asked the following questions: (1) “How many days a week are you generally exposed to passive smoking?”; (2) “How often are you exposed to passive smoking on these days?”; and (3) “How many years have you had this exposure to passive smoking?” The cumulative exposures to passive smoking were calculated according to the responses to these three questions, and the participants were further divided into three groups, by the 25th quartile (40 days) and the 75th quartile (300 days).

### 2.3. Assessment of Cognitive Function

The Chinese version of the Mini-Mental State Examination (CMMSE) was used to assess the cognitive function of the participants. CMMSE was translated from MMSE with full consideration of the Chinese language and culture by a bi-national team of psychiatrists and social scientists in the 1980s. It showed high validity in the identification of cognitive function, with diagnostic sensitivity of 80–90% and specificity of 70–80% [12,13]. It includes a total of 30 items, with a full score of 30, and higher scores indicating better cognitive function. The following education-specific cutoff values were used to define cognitive impairment: ≤17 points for those with no education, ≤20 for those with primary education only, and ≤24 for those with education beyond the primary level [12].

### 2.4. Covariates

The following participant information was collected: (1) demographic data, including age, sex, ethnic group, body mass index (BMI), education level (illiterate or semiliterate, primary school, junior high school, high school graduation or higher), marital status (unmarried, married, widowed, or divorced), job (never worked, farmer, housework, standard work, and other), and family income; (2) potential covariates of passive smoking exposure, including participation in group activities (never, occasionally, and frequently), smoking status (non-smokers, current smokers, and ex-smokers), alcohol consumption (non-drinkers, current drinkers, and ex-drinkers), tea consumption (non-drinkers, current drinkers, and ex-drinkers), physical exercise, or work. We used patient medical records to determine the presence of underlying diseases (stroke, high blood pressure, hyperlipidemia, diabetes, coronary heart disease, chronic bronchitis, gallstones, tumor, arthritis, cataracts, and others).

### 2.5. Statistical Analysis

We compared the distributions of demographic variables and other covariates between participants with and those without passive smoking exposure, using the t-test or Chi-square test. We used a multivariate regression model to calculate the adjusted relative risks (RRs) and 95% confidence intervals (CIs) of cognitive impairment among participants who were exposed to passive smoking. We further performed multivariate regression analyses to assess the relationship between passive smoking exposure and cognitive impairment among participants with and without active smoking. Statistical testing was conducted with a two-tailed α value of 0.05. All analyses were performed by using IBM SPSS Statistics for Windows, version 20.0 (IBM Corp., Armonk, NY, USA).

## 3. Results

Of the 7000 participants in this study, 993 (14.2%) were exposed to passive smoking in the living environment. Compared to those not exposed to passive smoking, these 993 participants were more likely to be women and of younger age; have never worked; have a higher family income; participate in group activities; be tea drinkers; and have underlying diseases, including chronic bronchitis, gallstones, arthritis, and cataract (Table 1). No significant differences in other factors were observed between the exposed and unexposed groups (Table 1).

Overall, 7000 participants with normal cognitive function at baseline were followed up for two years, of whom 1224 (17.5%) developed cognitive impairment. Table 2 shows the association between passive smoking exposure and cognitive impairment. After adjusting for covariates, including sex, age, body mass index (BMI), education, marital status, job type, family income, living or eating alone, participation in group activities, sleep quality, napping, alcohol consumption, tea consumption, water consumption, physical exercise, and work, the association between passive smoking exposure and cognitive impairment remained significant (RR: 1.16; 95% CI: 1.01–1.35). However, we did not observe a dose–response relationship between the cumulative dose of exposure (days) and cognitive impairment.

We further analyzed the relationship between passive smoking exposure and cognitive impairment among smoking and non-smoking participants (Table 3). The harmful effects of passive smoking on cognitive impairment were observed in non-active smokers (RR: 1.24; 95% CI: 1.06–1.46), but not in active smokers (RR: 1.11; 95% CI: 0.71–1.92). Among non-smokers, those exposed to passive smoking for a cumulative dose of 40–299 days had a higher risk of cognitive impairment than that in participants with exposures of less than 40 days (RR: 1.67; 95% CI: 1.00–2.80). However, no such harmful effect was observed in participants with cumulative doses over 300 days (Figure 1). An adjusted RR was not obtained for active smokers, because of the insufficient sample size.

## 4. Discussion

This prospective study assessed the relationship between passive smoking exposure in the living environment and the incidence of cognitive impairment among older adults. Our results revealed that passive smoking exposure exerted significant harmful effects on cognitive function, particularly in non-smokers.

Previous studies have shown that passive smoking can increase the risk of cognitive impairment or dementia, which is a possible outcome of cognitive impairment [14,15,16]. The results of our study also provided evidence of the harmful effect of passive smoking exposure on cognitive function. Passive smoking can damage the cardiovascular system by increasing platelet coagulability, leading to endothelial dysfunction [17,18]. Endothelial dysfunction might be related to the faulty clearance of amyloid beta-peptide across the blood-brain barrier, which plays an essential role in cognitive impairment [19]. Furthermore, the results of animal experiments have demonstrated that chronic smoking exposure can suppress synaptic function or cause other neuropathological changes, which might explain the early phases of neurodegeneration in brains [20]. Tobacco-specific procarcinogens may reduce neuronal mass in specific regions of the brain related to learning and memory [21]. A third important explanation is that the carbon monoxide (CO) in tobacco smoke interferes with the flow of oxygen through the blood to the brain, which may, in turn, impair cognitive function.

Similar to other studies, we also performed dose–response tests of passive smoking [22,23,24]. However, in contrast to previous results, we did not observe a risk trend between exposure dose and cognitive impairment. To assess the threshold effect of cumulative passive smoke exposure and to control for the possible short-term beneficial effects on cognitive function (especially on memory and attention) caused by nicotine [25], we filtered out participants who had passively smoked for more than 10 years before calculating and dividing the total exposure time and into four levels. Higher exposure to passive smoking did not show a significant risk effect on cognitive function. While several studies have shown no association between passive smoking and dementia, others have shown an inverse relationship between serum cotinine level and cognitive function; i.e., the effect was more significant at lower levels of exposure [26]. These conflicting results may be because of differences in the ventilation of the living environment of passive smokers, the actual effective dose of passive smoking exposure, insufficient sample size, recall bias from participants, and other factors related to individual differences.

The results of the independent analyses of participants who actively smoked and for those who did not revealed that the influence of the passive smoking on cognitive impairment was not significant in participants who were actively smoking but was significant in those who were not actively smoking. This finding suggests that passive smoking exposure was more harmful to non-active smokers than to active smokers. Previous studies have reported the negative effect of exposure to secondhand smoke on cognitive function among non-smokers [22,24]. One possible explanation is that nicotine reduces the activity of monoamine oxidase, which can cause nerve damage, allowing short-term cognitive improvements in people with cognitive impairment to mask some symptoms of the disease [27,28]. This makes it difficult to observe the harmful effects of passive smoking on the cognitive function in active smokers. In contrast, the cover-up effect is weaker in non-smokers, allowing the harmful effects of passive smoking to be observed in non-smokers. However, the different harmful effects and potential mechanisms of passive smoking on cognitive function between smokers and non-smokers have rarely been assessed. Further research into these differences is warranted, especially regarding genetic susceptibilities.

Nicotine may have some beneficial short-term effects on the cognitive function, particularly in areas related to memory and attention [25]. However, effects of smoking on the cognitive function is primarily observed in people with impaired cognitive function, particularly in those with neurological or psychiatric disorders and not in people with normal cognitive function. However, smoking tends to only improve the cognitive function in the short-term. Although nicotinoid nerve excitation mediated by nicotine receptors can improve the cognitive function, nicotine has many negative effects, such as damaging the blood vessels, increasing the oxidative stress, affecting mitochondrial energy metabolism, decreasing the function of synaptic network connectivity, and influencing metabolic enzymes related to A beta or tau protein. Therefore, in the long-term, smoking harms the cognitive function [29,30].

Our findings were based on a prospective design. This ensured a causal relationship between passive smoking exposure and cognitive impairment. We also compared the effects of passive smoking exposure on cognitive function among smokers and non-smokers, which has been rarely reported previously. Nevertheless, this study has several limitations. First, investigations based on self-reports may lead to overestimation or underestimation of the passive smoking exposure [31,32], which might affect the accuracy of the harmful effects of this exposure on cognitive impairment. The multiple regression analysis also suggested that the association between passive smoking and cognitive impairment may be weak, because at least ten confounding factors influence the detrimental effect. The other limitation is that our study only calculated passive smoking exposure time in living environments. Other possible exposures include unconscious exposure in public spaces. Some studies using serum cotinine as a biomarker for passive smoking exposure [26,33] have reported its significant negative association with cognitive performance. In contrast, urine sampling is much less invasive and can also be used to quantify cotinine levels [34]. Continuous long-term monitoring of urine or serum nicotine levels and epidemiological investigation are required to objectively assess passive smoking exposure.

## 5. Conclusions

Our findings appear to indicate that there is an association between passive smoking exposure in the living environment and an increased risk of cognitive impairment among older adults, with a greater harmful effect in non-smokers compared to that in smokers. Considering that more than 90% of the world’s population is not completely protected by smoke-free public health regulations [35], we should appeal to families and societies to reduce the exposure to passive smoking, especially for older adults. The Chinese government has taken strict measures to limit smoking in public places in recent years and has made great progress. However, smokers may increase the amount of smoke in the family household and other living environments, which increases the exposure of the co-dwellers to passive smoking. In light of this study, we suggest that the government establish a community supervision mechanism and take the family as a unit to persuade smokers to quit or control smoking. In addition, further studies are required to assess the dose–response relationship between passive smoking exposure and cognitive impairment and to determine the potential mechanisms in smokers and non-smokers.

## Figures and Tables

**Figure 1 ijerph-17-01402-f001:**
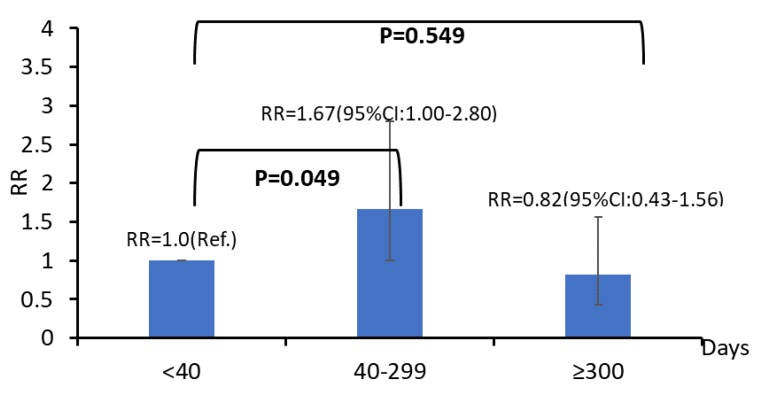
Relationship between the cumulative dose (days) of passive smoking exposure and cognitive impairment among non-smokers (ZJMPHS Program, Zhejiang, China). Adjusted for age, sex, ethnicity, body mass index, education, marital status, job type, family income, participation in group activities, smoking, alcohol consumption, tea consumption, physical exercise, and work. RR: relative risk; CI: confidence interval; ZJMPHS Program: Zhejiang Major Public Health Surveillance Program.

**Table 1 ijerph-17-01402-t001:** Participant’s characteristics according to passive smoking status (ZJMPHS Program, Zhejiang, China).

Variables	Passive Smoking Exposure				*p*-Value
	No		Yes		
	n	%	n	%	
Age (years)					
Mean (SD)	70.3	(7.0)	69.0	(6.5)	<0.001
Ethnicity					
Han	5794	86.4	914	13.6	<0.001
Other	172	68.5	79	31.5	
Sex					
Female	2932	82.1	641	17.9	<0.001
Male	3035	89.6	352	10.4	
BMI, kg/m^2^					
18.5–24.99	3740	86.4	589	13.6	0.434
<18.5	299	87.7	42	12.3	
>24.99	1663	85.5	283	14.5	
Education					
Illiterate or semiliterate	2902	85.8	482	14.2	0.448
Primary school	2616	85.3	450	14.7	
Junior high school	380	88.0	52	12.0	
High school graduation or higher	69	88.5	9	11.5	
Marital status					
Unmarried	80	83.3	16	16.7	<0.001
Married	4637	84.6	847	15.4	
Widowed	1213	90.7	125	9.3	
Divorced	25	86.2	4	13.8	
Job					
Never worked	1356	80.8	322	19.2	<0.001
Farmers	2697	88.2	361	11.8	
Housework	717	87.3	104	12.7	
Workers	616	84.5	113	15.5	
Others	557	86.2	89	13.8	
Family income (1000 ¥/year)					
<10	952	90.4	101	9.6	<0.001
10–19	1350	88.6	174	11.4	
20–49	1636	85.6	276	14.4	
50–99	992	81.4	226	18.6	
≥100	1032	82.8	214	17.2	
Participation in group activities					
Never	3706	86.8	563	13.2	0.003
Occasional	1522	83.6	298	16.4	
Frequent	735	84.8	132	15.2	
Smoking					
Non-smokers	4211	85.4	718	14.6	0.284
Current smokers	1164	85.8	192	14.2	
Ex-smokers	592	87.7	83	12.3	
Alcohol consumption					
Nondrinkers	4012	85.9	656	14.1	0.076
Current drinkers	1569	86.1	253	13.9	
Ex-drinkers	378	82.2	82	17.8	
Tea drinking					
Nondrinkers	4389	87.2	646	12.8	<0.001
Current drinkers	1453	82.0	319	18.0	
Ex-drinkers	110	84.6	20	15.4	
Physical exercise					
No	4824	86.1	776	13.9	0.064
Yes	1139	84.2	214	15.8	
Physical work					
No	3975	86.4	628	13.6	0.112
Yes	1981	84.9	351	15.1	
Stroke					
No	5807	85.7	971	14.3	0.314
Yes	159	88.3	21	11.7	
High blood pressure					
No	3140	85.2	546	14.8	0.169
Yes	2826	86.3	447	13.7	
Hyperlipidemia					
No	5710	85.8	946	14.2	0.437
Yes	250	84.2	47	15.8	
Diabetes					
No	5384	85.9	887	14.1	0.329
Yes	577	84.5	106	15.5	
Coronary heart disease					
No	5743	85.7	960	14.3	0.552
Yes	221	87.0	33	13.0	
Chronic bronchitis					
No	5851	85.9	959	14.1	0.003
Yes	112	77.2	33	22.8	
Gallstones					
No	5774	86.0	940	14.0	0.001
Yes	190	78.2	53	21.8	
Tumor					
No	5823	85.7	973	14.3	0.497
Yes	141	87.6	20	12.4	
Arthritis					
No	5768	85.9	946	14.1	0.017
Yes	188	80.3	46	19.7	
Cataract					
No	5687	85.9	930	14.1	0.020
Yes	276	81.4	63	18.6	
Depressive symptoms					
Normal	5157	85.6	869	14.4	0.524
Mild depression	607	86.3	96	13.7	
Moderate depression	164	86.8	25	13.2	
Heavy depression	39	92.9	3	7.1	

ZJMPHS Program: Zhejiang Major Public Health Surveillance Program.

**Table 2 ijerph-17-01402-t002:** Adjusted relative risks (RRs) for the association between passive smoking exposure and cognitive impairment (ZJMPHS Program, Zhejiang, China).

Covariates	Cognitive Impairment				Multivariate Adjusted Regression Analyses	
	No		Yes			
	n	%	n	%	Adjusted RR (95% CI) ^a^	*p*-Value
Age (years)						
Mean (SD)	69.4	(6.5)	73.3	(7.8)	1.04 (1.03–1.05)	<0.001
Male sex	2883	85.1	504	14.9	0.85 (0.73–0.969)	0.040
Han ethnicity	5486	81.8	1222	18.2	1.30 (0.92–1.84)	0.141
BMI, kg/m^2^						
18.5–24.99	3565	82.4	764	17.6	1.00	
<18.50	241	70.7	100	29.3	1.21 (1.05–1.39)	0.007
>24.99	1635	84.0	311	16.0	0.94 (0.84–1.06)	0.305
Education						
Illiterate or semiliterate	2647	78.2	737	21.8	1.00	
Primary school	2612	85.2	454	14.8	0.99 (0.89–1.12)	0.964
Junior high school	372	86.1	60	13.9	1.03 (0.80–1.33)	0.825
High school or higher	69	88.5	9	11.5	0.86 (0.49–1.51)	0.605
Marital status						
Married	83	86.5	13	13.5	1.00	
Unmarried	4599	83.9	885	16.1	1.31 (0.77–2.25)	0.319
Widowed	986	73.7	352	26.3	1.27 (0.74–2.19)	0.390
Divorced	23	79.3	6	20.7	1.56 (0.57–4.24)	0.388
Job						
Never worked	1254	74.7	424	25.3	1.00	
Farmers	2545	83.2	513	16.8	0.88 (0.78–0.99)	0.035
Housework	657	80.0	164	20.0	0.83 (0.71–0.98)	0.023
Workers	642	88.1	87	11.9	0.78 (0.62–0.99)	0.038
Others	577	89.3	69	10.7	0.77 (0.60–0.99)	0.043
Family income (1000 ¥/year)						
<10	734	69.7	319	30.3	1.00	
10–19	1175	77.1	349	22.9	0.93 (0.82–1.06)	0.273
20–49	1673	87.5	239	12.5	0.62 (0.52–0.73)	<0.001
50–99	1026	84.2	192	15.8	0.74 (0.62–0.87)	<0.001
≥100	1087	87.2	159	12.8	0.58 (0.48–0.69)	<0.001
Participation in group activities						
Never	3359	78.7	910	21.3	1.00	
Occasionally	1559	85.7	261	14.3	0.75 (0.66–0.85)	<0.001
Frequently	778	89.7	89	10.3	0.56 (0.46–0.69)	<0.001
Smoking						
Non-smokers	3951	80.2	978	19.8	1.00	
Current smokers	1163	85.8	193	14.2	1.01 (0.84–1.21)	0.923
Ex-smokers	586	86.8	89	13.2	0.88 (0.69–1.12)	0.295
Alcohol consumption						
Non-drinkers	3751	80.4	917	19.6	1.00	
Current drinkers	1554	85.3	268	14.7	1.13 (0.97–1.30)	0.109
Ex-drinkers	389	84.6	71	15.4	0.95 (0.74–1.20)	0.648
Tea consumption						
Non-drinkers	4032	80.1	1003	19.9	1.00	
Current drinkers	1534	86.6	238	13.4	0.81 (0.70–0.94)	0.004
Ex-drinkers	117	90.0	13	10.0	0.49 (0.27–0.88)	0.016
Physical exercise	1198	88.5	155	11.5	0.68 (0.58–0.81)	<0.001
Physical work	2083	89.3	249	10.7	0.70 (0.61–0.81)	<0.001
Passive smoking exposure	819	82.5	174	17.5	1.16 (1.01–1.35)	0.047
Cumulative dose of passive smoking exposure (days)						
<40	174	81.3	40	18.7	1.00	
40–299	336	80.8	80	19.2	1.17 (0.62–1.46)	0.277
≥300	199	87.3	29	12.7	0.72 (0.41–1.32)	0.421

^a^ Adjusted for age, sex, ethnicity, body mass index (BMI), education, marital status, job type, family income, participation in group activities, smoking, alcohol consumption, tea consumption, physical exercise, or work. RR: relative risk; CI: confidence interval; ZJMPHS Program: Zhejiang Major Public Health Surveillance Program.

**Table 3 ijerph-17-01402-t003:** Stratified analysis of the relationship between passive smoking exposure and cognitive impairment with respect to active smoking (ZJMPHS Program, Zhejiang, China).

Active Smoking	Passive Smoking Exposure	Cognitive Impairment				Multivariate Adjusted Regression Analyses	
		No		Yes			
		n	%	n	%	Adjusted RR ^a^ (95% CI)	*p*-Value
No	No	3884	80.9	919	19.1	1.00	
	Yes	653	81.5	148	18.5	1.24 (1.06–1.46)	0.008
Yes	No	997	85.7	167	14.3	1.00	
	Yes	166	86.5	26	13.5	1.11 (0.71–1.92)	0.610

^a^ Adjusted for age, sex, ethnicity, body mass index (BMI), education, marital status, job type, family income, participation in group activities, smoking, alcohol consumption, tea consumption, physical exercise, or work. RR: relative risk; CI: confidence interval; ZJMPHS Program: Zhejiang Major Public Health Surveillance Program.

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
