# Peer review of "Passive Smoking Exposure in Living Environments Reduces Cognitive Function: A Prospective Cohort Study in Older Adults"

_ijerph, 2020, doi:10.3390/ijerph17041402_

Round 1

Reviewer 1 Report

The article is well written and the exposure of the content is clear and effective. In my opinion, authors must also consider other aspects in their research: 1. coffee/tea consumption: considering the known effect of caffeine on cognitive function, also the assessment of coffee consumption could have been useful. Moreover, this informations could have been integrated with tea consumption, to have an overall approximation of nervine beverages consumption.  2. cotinine measurements:  the assessment of cotinine in the biological samples can be assessed effectively not only in serum but also in urine (see, for example, this work: Bono R., 2019, Bisphenol a, tobacco smoke, and age as predictors of oxidative stress in children and adolescents). In this regard, collecting urine is easier and less invasive than blood sampling.   3. graphs: finally, I suggest to improve statistical analyses with 1 or 2 graphs, to better explain the results.

Author Response

Point 1: coffee/tea consumption: considering the known effect of caffeine on cognitive function, also the assessment of coffee consumption could have been useful. Moreover, this information could have been integrated with tea consumption, to have an overall approximation of nervine beverages consumption.

Response1: Thank you very much for your suggestion. Considering the older adults had little coffee consumption in China, especially in rural or suburban areas, we didn’t investigate the coffee consumption of participants. A case-control study of “Tea, coffee, and head and neck cancer risk in a multicenter study in east Asia” conducted in eight provinces of China showed the mean coffee intake was only 0.7 cups (about 166 ml) per week among 65-85 years old people[1]. However, coffee will become increasingly popular among the older adults in the coming years. Therefore, we will investigate coffee consumption in the following follow-up and assess its effect on cognitive impairment. Thanks again for your advice.

References:

  1. Shuang Li, Yuan-Chin Amy Lee, Qian Li, etc. Tea, coffee, and head and neck cancer risk in a multicenter study in east Asia. Oral Cancer. 2018, 2,(2), 57-65.

Point 2:  cotinine measurements: the assessment of cotinine in the biological samples can be assessed effectively not only in serum but also in urine (see, for example, this work: Bono R., 2019, Bisphenol a, tobacco smoke, and age as predictors of oxidative stress in children and adolescents). In this regard, collecting urine is easier and less invasive than blood sampling.

Response 2: Thank you very much for your suggestion. A shift to quantitative measurement of cotinine levels in urine would make the study much easier to implement and much less invasive. Your suggestion will greatly improve our study design and make the results more reliable. We have revised the discussion part according to your suggestion.(please refer to lines 195-197.)

Point 3: graphs: finally, I suggest to improve statistical analyses with 1 or 2 graphs, to better explain the results.

Response3: Thank you very much for your suggestion. we have made a graph to illustrate the relationship between cumulative dose(days) of passive smoking exposure and cognitive impairment among non-smokers.(please refer to figure 1).

Reviewer 2 Report

The authors present observations from the Zhejiang Province in China.  Fan et al report that a statistically significant adjusted relative risk of cognitive impairment among passive smokers but not active smokers.  There are major concerns regarding this manuscript:

1) The authors use the Chinese version of the Mini-Mental State Examination to assess cognitive function.  They should describe the psychometric validity of the Questionnaire.  It is not clear whether the questionnaire is valid to assess the cognitive function in the population being studied.

2) The lack of a dose-response within the passive smoking group and the lack of significant response in active smokers brings to question the conclusions from this analysis.  The authors do not provide compelling rationale for the conclusions that "passive smoking exposure increased the risk of cognitive impairment in older adults, especially non-smokers."

3) The statistically significant impact for the confounders in the multivariate regression analysis e.g. age, gender and close to 17 variables as illustrated in Table 2 suggests that the strength of the association is very weak at best and the authors should indicate as such.

4) The authors do not provide the appropriate scientific explanation for the possible association, particularly given the many reports of enhanced cognitive function due to nicotine.  

Author Response

Point 1: The authors use the Chinese version of the Mini-Mental State Examination to assess cognitive function.  They should describe the psychometric validity of the Questionnaire.  It is not clear whether the questionnaire is valid to assess the cognitive function in the population being studied.

Response 1: Thank you very much for your comments. Chinese version of MMSE (CMMSE) was translated from MMSE with full consideration of the Chinese language and culture by a bi-national team of psychiatrists and social scientists in the 1980s. It showed high validity in the identification of cognitive function, with diagnostic sensitivity of 80–90% and specificity of 70–80% [1,2]. At present, CMMSE is widely used in the detection of cognitive impairment and is used as the gold standard for the development of new scales. Based on your suggestion, we have supplemented the description in the methods section.

References:

  1. Wang, Z.Y.; Zhang, M.Y.; Zhai, G.Y.; Chen, J.X.; Zhao, J. Application of the Chinese version of the Mini-Mental State Examination. Shanghai Arch Psychiatr 1989, 7(3), 108–111.
  2. Zhang, Z.J. Handbook of behavioral medical scales. Shandong, China: Chinese Journal of Behavioral Medical Science 2005.

Point 2: The lack of a dose-response within the passive smoking group and the lack of significant response in active smokers brings to question the conclusions from this analysis.  The authors do not provide compelling rationale for the conclusions that "passive smoking exposure increased the risk of cognitive impairment in older adults, especially non-smokers."

Response 2: Thank you very much for your comments. One possible explanation is that nicotine reduces the activity of monoamine oxidase, which can cause nerve damage, allowing short-term cognitive improvements in people with cognitive impairment to mask some symptoms of the disease [1, 2]. This makes it difficult to observe the harmful effects of passive smoking on the cognitive function in active smokers. In contrast, the cover-up effect is weaker in non-smokers, allowing the harmful effects of passive smoking to be observed in non-smokers. We have revised the discussion section (please refer to lines 165-170).

References:

  1. Murray KN1, Abeles N. Nicotine's effect on neural and cognitive functioning in an aging population. Aging Ment Health. 2002, 6(2):129-38.
  2. Teaktong, A. J. Graham, M. Johnson, J. A. Court, and E. K. Perry, “Selective changes in nicotinic acetylcholine receptor subtypes related to tobacco smoking: an immunohistochemical study,” Neuropathology and Applied Neurobiology. 2004, 30 (3): 243-254.

Point 3: The statistically significant impact for the confounders in the multivariate regression analysis e.g. age, gender and close to 17 variables as illustrated in Table 2 suggests that the strength of the association is very weak at best and the authors should indicate as such.

Response 3 : Thank you very much for your comments. We have added the following description to the discussion section(please refer to lines 189-192):  The multiple regression analysis also suggested that the association between passive smoking and cognitive impairment may be weak because at least ten confounding factors influence the detrimental effect.

Point 4: The authors do not provide the appropriate scientific explanation for the possible association, particularly given the many reports of enhanced cognitive function due to nicotine.

Response 4: Thank you very much for your comments. Nicotine may have some beneficial short-term effects on the cognitive function, particularly in areas related to memory and attention [1]. However, effects of smoking on the cognitive function is primarily observed in people with impaired cognitive function, particularly in those with neurological or psychiatric disorders and not in people with normal cognitive function. However, smoking tends to only improve the cognitive function in the short term. Although nicotinoid nerve excitation mediated by nicotine receptors can improve the cognitive function, nicotine has many negative effects such as damaging the blood vessels, increasing the oxidative stress, affecting mitochondrial energy metabolism, decreasing the function of synaptic network connectivity, and influencing metabolic enzymes related to A beta or tau protein. Therefore, in the long term, smoking harms the cognitive function [2,3]. We have revised the discussion section.(please refer to lines 174-183).

References:

  1. Newhouse, P.A.; Potter, A.; Singh, A. Effects of nicotinic stimulation on cognitive performance. Curr Opin Pharmacol 2004, 4, 36-46.
  2. Cervilla, J.A.; Prince, M.; Mann, A. Smoking, drinking, and incident cognitive impairment: a cohort community based study included in the Gospel Oak project. J Neurol Neurosurg Psychiatry 2000, 68, 622-626.
  3. Hill, R.D.; Nilsson, L.G.; Nyberg, L.; Backman, L. Cigarette smoking and cognitive performance in healthy Swedish adults. Age Ageing 2003, 32 , 548-550.

Reviewer 3 Report

In the conclusions, the authors state the following:
“Our findings demonstrated an association between passive smoking exposure in the living environment and an increased risk of cognitive impairment among older adults, with a greater harmful effect in non-smokers compared to that in smokers. Considering that more than 90% of the 179 world’s population is not completely protected by smoke-free public health regulations [29], we should appeal to families and societies to reduce the exposure to passive smoking, especially for older adults. However, further studies are required to assess the dose-response relationship between passive smoking exposure and cognitive impairment and to determine the potential mechanisms in smokers and non-smokers. ”
In view of the results obtained, they should not affirm so conclusive that "our findings demonstrated an association between passive smoking exposure in the living environment and an increased risk of cognitive impairment among older adults". The word “demonstrate”, it is very strong. Replace with "appear to indicate that there is an association".
They should also be more objective in their suggestions for controlling adult exposure to reducing the exposure to passive smoking

Author Response

Point 1: In view of the results obtained, they should not affirm so conclusive that "our findings demonstrated an association between passive smoking exposure in the living environment and an increased risk of cognitive impairment among older adults". The word “demonstrate”, it is very strong. Replace with "appear to indicate that there is an association".

Response 1: Thank you very much for your suggestions. We have modified the less accurate words with “appear to indicate that there is an association”.

Point 2 They should also be more objective in their suggestions for controlling adult exposure to reducing the exposure to passive smoking

Response 2: Thank you very much for your suggestions. The Chinese government has taken strict measures to limit smoking in public places in recent years and has made great progress. However, smokers may increase the amount of smoke in the family household and other living environments, which increases the exposure of the co-dwellers to passive smoking. In light of this study, we suggest that the government establish a community supervision mechanism and take the family as a unit to persuade smokers to quit or control smoking. We have revised the suggestions (please refer to lines 205-209).

Round 2

Reviewer 2 Report

The authors have adequately addressed my comments